# A dataset of daily near-surface air temperature in China from 1979 to 2018

**Shu Fang [1,2]★, Kebiao Mao[1,3]★, Xueqi Xia[2], Ping Wang[1,4], Jiancheng Shi[5], Sayed M. Bateni[6], Tongren Xu[7], Mengmeng Cao[1], Essam Heggy[8,9]**

1.Institute of agricultural resources and regional planning, Chinese Academy of Agricultural Sciences, Beijing, 100081, China.

2. School of Earth Sciences and Resources, China University of Geosciences, Beijing, 100083, China. xiaxueqi@cugb.edu.cn

3. School of Physics and Electronic-Engineering, Ningxia University, Yinchuan 750021, China.

4. School of Surveying and Geo-Informatics, Shandong Jianzhu University, Jinan, 250100, China.

5. National Space Science Center, Chinese Academy of Sciences, Beijing, 100190, China. shijiancheng@nssc.ac.cn

6. Department of Civil and Environmental Engineering and Water Resources Research Center, University of Hawaii at Manoa, Honolulu, HI 96822, USA; smbateni@hawaii.edu

7. State Key Laboratory of Earth Surface Processes and Resource Ecology, School of Natural Resources, Faculty of Geographical Science, Beijing Normal University, Beijing 100875, China; xutr@bnu.edu.cn

8. Viterbi School of Engineering, University of Southern California, Los Angeles, CA 90089, USA; heggy@usc.edu

9. Jet Propulsion Laboratory, California Institute of Technology, Pasadena, CA 91109, USA. Correspondence to: Kebiao Mao (maokebiao@caas.cn)

★ These authors contributed equally to this work.

**Abstract:** $T_a$ (Near-surface air temperature) is an important physical parameter that reflects

climate change. Although there are currently many methods to obtain the daily maximum ($T_{max}$), minimum ($T_{min}$), and average ($T_{avg}$) temperature (meteorological stations, remote sensing, and reanalysis data), these methods are affected by multiple factors. In order to obtain daily $T_a$ data ($T_{max}$, $T_{min}$, and $T_{avg}$) with high spatial and temporal resolution in China, we fully analyzed the advantages and disadvantages of various existing data (reanalysis, remote sensing, and in situ

data). Different $T_a$ reconstruction models are constructed for different weather conditions, and we further improve data accuracy through building correction equations for different regions. Finally, a dataset of daily temperature ($T_{max}$, $T_{min}$, and $T_{avg}$) in China from 1979 to 2018 was obtained with a spatial resolution of 0.1°. For $T_{max}$, validation using in situ data shows that the root mean square error (RMSE) ranges from 0.86 ℃ to 1.78 ℃, the mean absolute error (MAE) varies from 0.63 ℃



to 1.40 °C, and the Pearson coefficient ($R^2$) ranges from 0.96 to 0.99. For $T_{min}$, RMSE ranges from

0.78 °C to 2.09 °C, the MAE varies from 0.58 °C to 1.61 °C, and the $R^2$ ranges from 0.95 to 0.99.

For $T_{avg}$, RMSE ranges from 0.35 °C to 1.00 °C, the MAE varies from 0.27 °C to 0.68 °C, and the

$R^2$ ranges from 0.99 to 1.00. Furthermore, a variety of evaluation indicators were used to analyze

the temporal and spatial variation trends of $T_a$, and the $T_{avg}$ increase was more than 0.03 °C/a,

which are consistent with the general global warming trend. In conclusion, this dataset had a high

spatial resolution and reliable accuracy, which makes up for the previous missing temperature

value ($T_{max}$, $T_{min}$, and $T_{avg}$) at high spatial resolution. This dataset also provides key parameters

for the study of climate change, especially high-temperature drought and low-temperature chilling

damage,        which        is        publicly        available        with        the        following        DOI:

https://doi.org/10.5281/zenodo.5502275 (Fang et al., 2021).

## 1. Introduction

$T_a$ (Near-surface air temperature) is an important variable that reflects global climate change, and

it significantly affects the cyclical conversion of energy and matter in all spheres of the earth (Gao

et al., 2012, 2014). Obtaining accurate grid air temperature is helpful for research on urban heat

island effects, the ecological environment changes, vegetation phenology development, crop yield

fluctuation, and energy dynamic balance (Lin et al., 2012; Bolstad et al., 1998). In this study, $T_a$

refers to the daily maximum ($T_{max}$), minimum ($T_{min}$), and average temperatures ($T_{avg}$) of daily

near-surface air temperature, which are important input parameters for hydrological,

environmental, and crop models (Han et al., 2020; He et al., 2020; Mostovoy et al., 2006; Schaer

et al., 2004). They can accurately reflect the frequency and extent of the occurrence and

development of extreme climate events (Zhang et al., 2017; Miao et al., 2016). With the increase

in global warming, the temperature gradually increases and the extremely cold days and nights

gradually shorten (Ding et al., 2006; Liao et al., 2020). However, the intensity and duration of

extreme weather events are also increasing, and continuous bad weather in some years leads to

frequent meteorological disasters (Ryoo et al., 2010). China is a country where extreme weather

events frequently occur, which causes huge economic losses (Kharin et al., 2007; Kong et al.,

2020). Therefore, it is essential to obtain the spatio-temporal changes of $T_a$ for studying extreme

weather events, meteorological disasters leading to agricultural production reduction.



$T_a$ is affected by many factors of the earth's system, resulting in frequent and complicated daily

temperature fluctuations (Schwingshackl et al., 2018; Chen et al., 2014). At present, $T_a$ is obtained

mainly through three methods: monitoring $T_a$ via meteorological stations, estimating $T_a$ from $T_s$

(land surface temperature) retrieved from remote sensing, and obtaining $T_a$ through the

assimilation model. The temperature with high time resolution can be obtained through the

measurement of the meteorological station, which can avoid the influence of clouds and rain,

preserving good data integrity, continuity, and accuracy. However, the number of meteorological

stations is limited and unevenly distributed, especially for mountainous regions (Mao et al., 2008;

Gao et al., 2018; Zhao et al., 2020). Most meteorological stations are located in sparsely populated

areas far away from cities and cannot accurately monitor changes in urban temperature caused by

the urban heat island effect (He and Wang, 2020). Moreover, owing to the aging of meteorological

station equipment, the observation data may be incomplete. Although many interpolation methods,

such as Kriging, Cubic Spline, and Inverse Distance Weight interpolations are available, the

difference in density between stations has some impact on the interpolation accuracy (Tang et al.,

2020; Tomasz et al., 2016; Tencer et al., 2011).

Satellite sensors can provide global coverage and high spatial resolution data, which can be

used to estimate $T_a$. The estimation methods are mainly divided into five categories. The first

method is the statistical regression method, which simulates the fluctuation of daily temperature

by establishing a regression model between temperature and other parameters (Wen et al., 2020).

The model parameters mainly include altitude, latitude and longitude, solar phase angle, and day

length (Zhu et al., 2013; Zhang et al., 2015). The second method is the temperature vegetation

index (TVX) method, which is a method for air temperature estimation based on the negative

correlation between surface temperature and vegetation index (Xing et al., 2020). The third

method is the energy balance method. It is generally considered that the sum of the net radiation

and anthropogenic heat flux in the surface energy is equal to the sum of the surface sensible heat

flux and latent heat flux to calculate the surface air temperature (Benali et al., 2012). The fourth

method is the atmospheric temperature profile extrapolation method, which uses the vertical

attenuation rate obtained from the atmospheric temperature profile to calculate the $T_a$ (Wen et al.,

2020). The fifth method is a machine learning method that uses polynomial regression or neural

network algorithms to improve $T_a$ estimation errors (Mao et al, 2008; Wen et al., 2020). Sensors are susceptible to weather phenomena, such as clouds and rain, leading to missing data or reduced quality. In addition, these methods of inferring $T_a$ are mostly suitable for clear sky conditions, which still need to be further expanded to establish an estimation model of $T_s$ to $T_a$ under different weather conditions.

In recent years, the reanalysis data generated by the global assimilation model has provided many datasets of geophysical parameters, including near-surface temperature, which overcome most of the above-mentioned problems caused by abnormal weather. The NCEP/NCAR reanalysis dataset was developed by the National Center for Environmental Prediction and the National Center for Atmospheric Research (1948.1–2021.9), with a time resolution of 6 h and a spatial resolution of 2.5° (Kobayashi et al., 2015). The ERA5 dataset was released by the European Center for Medium-Range Weather Forecast (ECMWF; 1950.1–2021.9), with a time resolution of 1 h, and a spatial resolution of 0.3° (Hersbach et al., 2020; Dee et al., 2011; Taszarek et al., 2021; Lei et al., 2020). The Princeton Forcing surface model dataset was developed by Princeton University (1948.1–2006.12), with a time resolution of 3 h and a spatial resolution of 1.0° (Deng et al., 2010). To improve the accuracy of regional data, some researchers have developed different types of forcing datasets for the Chinese region. The representative dataset is the China Meteorological Forcing Dataset (CMFD) released by the Institute of Tibetan Plateau Research, Chinese Academy of Sciences (1979.1–2018.12), with a time resolution of 3 h and a spatial resolution of 0.1° (He et al., 2010; Yang et al., 2010; Yang and He, 2019). However, the dataset does not provide daily maximum and minimum temperatures. The grid dataset of daily surface temperature in China (V2.0, CMA) was released by the China Meteorological Administration (1961.1–2021.9), with a spatial resolution of 0.5°. This dataset only includes the daily maximum, minimum, and average temperatures, and its spatial resolution is low and the accuracy of local areas needs to be further improved. Although reanalysis datasets can obtain global near surface air temperature data, there is a lack of $T_{max}$, $T_{min}$ and $T_{avg}$ dataset with high spatial resolution and high precision.

In order to obtain a long-term $T_a$ ($T_{max}$, $T_{min}$, and $T_{avg}$) dataset with high spatial resolution in China based on the current reanalysis, remote sensing, and in situ data. We first analyze the
advantages and disadvantages of various existing data (reanalysis, remote sensing, in situ data, etc.). Then, different daily $T_a$ reconstruction models are constructed for different weather conditions. It makes up for the previous methods which are most suitable for clear sky conditions

and the insufficient estimation of all-weather conditions. We further improve data accuracy by building correction equations for different regions. Finally, a dataset of daily $T_a$ ($T_{max}$, $T_{min}$, and $T_{avg}$) in China from 1979 to 2018 was obtained with a spatial resolution of 0.1°. The comparison with in situ data and the existing reanalysis dataset is made.

## 2. Study area

China has a vast territory, with great undulations on the earth's surface, and a wide range of climate changes. In order to improve the accuracy of $T_a$ estimation, we divide China into six subregions shown in Figure 1 based on geographic location, altitude, rainfall, vegetation types and other natural environmental conditions. (I) The Northeastern Region is mainly including northeast China, which is located to the east of the Greater Khingan Range. This region is located

in the temperate monsoon climate zone, the annual precipitation is 400–1000 mm and cumulative temperature is between 2500°C and 4000°C. (II) The North China region is located in the area north of the Qinling-Huaihe River and south of the Inner Mongolia Plateau. This region is mostly located in the temperate monsoon climate zone, the annual accumulated temperature is between 3000°C and 4500°C, with hot and rainy summers and cold and dry winters. (III) The Central

Southern region is located south of the Qinling-Huaihe River and north of the tropical monsoon climate type. This region is located in the subtropical monsoon climate zone, the annual accumulated temperature is between 4500°C and 8000°C and the precipitation is mostly between 800 mm and 1600 mm. (IV) The Southern region is south of the Tropic of Cancer. This region is located in the tropical monsoon climate zone, the annual accumulated temperature is greater than

800°C, the annual minimum temperature is not less than 0°C, and there is no frost throughout the year. The annual precipitation mostly ranges from 1500 mm to 2000 mm. (V) The Northwest region is mainly distributed in the inland areas above 40° N latitude of China, located in the northwest of the Greater Khingan Range-Yin Shan-Ho–lan Mountains-Qilian Mountains line. It is far from the coast, water vapor transport is limited, and the annual precipitation is between 300

mm and 500 mm. Both the daily and the annual temperature differences are large, including

temperate desert, temperate grassy, and sub-frigid coniferous climates. (VI) The Qinghai–Tibet

Plateau region mainly includes the Qinghai-Tibet Plateau, the Andes Mountains, Mount Everest,

and other areas. This region is located in the plateau and mountainous climate zone, the annual

accumulated temperature is lower than 2000°C, the daily temperature range is large, and the

annual temperature range is small. This region has strong solar radiation, sufficient sunshine, and

little precipitation.

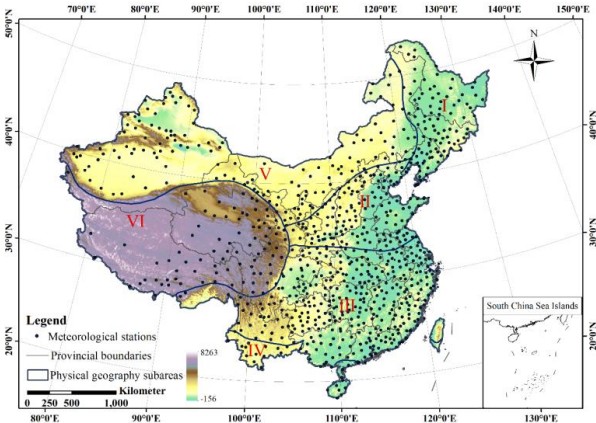

Figure 1. Scope map of the total study area and the six subregions. The black dots indicate the distribution

locations of meteorological stations; blue frame lines indicate the sub-study area range, represented by I, II, III,

IV, V, and VI.

## 3.   Data

### 3.1 Reanalysis data

The reanalysis dataset contains driving factors of surface elements in a large area, which can

provide highly complementary information and avoid data gaps and low pixel quality caused by

abnormal weather conditions. This study primarily used the CMFD and ERA5 datasets as the

reanalysis data sources.

CMFD data are a set of meteorological forcing datasets developed by the Institute of Tibetan

Plateau Research, Chinese Academy of Sciences (He et al., 2020; Yang et al., 2010; Yang and He,

2019). They are mainly based on the Global Land Data Assimilation System (GLDAS) as a

background dataset, using empirical knowledge algorithms and combining GLDAS with

measured data to obtain temperature data with a spatial resolution of 0.1°. The CMFD dataset

contains seven variables: 2-m air temperature, surface pressure, specific humidity, 10-m wind



speed, downward shortwave, and downward longwave radiation, and precipitation rate. The CMFD dataset covers the period from January 1979 to December 2018. In total, four types of time resolution products are provided every 3 h, daily, monthly and annual averages. At present, CMFD data are a comprehensive dataset with the longest regional time series and the highest spatial resolution in China. Many studies and analyses show that the dataset's accuracy is high enough to meet the application requirements (Zhang et al., 2019; Wang et al., 2017). Therefore, we use the 3-h temperature and daily temperature data of the CMFD to construct the $T_a$ model and make evaluation with this product, respectively. CMFD dataset is available through the China National Qinghai-Tibet Plateau Science Data Center (http://data.tpdc.ac.cn/zh-hans/data/8028b944-daaa-4511-8769-965612652c49/, last access: 1 November 2020).

ERA5 data is the fifth-generation product of atmospheric reanalysis global climate data launched by the ECMWF, replacing the ERA-Interim reanalysis data that was discontinued on August 31, 2019 (https://cds.climate.copernicus.eu/cdsapp#!/search?type=dataset&text=ERA5, last access: 1 December 2020). ERA5 data is generated based on the Cy41r2 model of the integrated forecasting system which has benefited from the development of data assimilation, model simulation, and model physics in recent years, and is generated by absorbing more ground monitoring, aircraft weather observation, and radio detection data. Compared with ERA-Interim data, ERA5 was significantly improved, such as higher temporal and spatial resolution, more vertical mode levels, and added other parameter products. ERA5 provides timely and updated quality checks on the data, which is convenient for providing stable, real-time, and long-term climate information. ERA5 includes many meteorological elements, including 2-m air temperature, 2-m relative humidity, sea level pressure, sea surface temperature, and precipitation. Since the release of ERA5 reanalysis data, many researchers have tested its applicability and accuracy. The results show that the accuracy of the ERA5 is better than the ERA-Interim data, and the higher temporal and spatial resolutions are conducive to the precise description of regional atmospheres. The details of these improvements are convenient for studying changes in small-scale atmospheric environments (Meng et al., 2018; Mo et al., 2021; Hillebrand et al., 2021). Therefore, the temperature data in the ERA5 data is selected to reconstruct the $T_a$ dataset.

## 3.2 Meteorological station data

The meteorological station data from 1979-2018 were used in this study were employed to build a $T_a$ model and make evaluations for existing datasets and new products. The measured data of meteorological stations are obtained from China National Meteorological Information Center

(http://www.nmic.cn/site/index.html, last access: 1 November 2020), including the daily temperature data of China's surface climate ($T_{max}$, $T_{min}$, and $T_{avg}$), hourly air temperature, and land surface temperature data. In order to further improve the data quality, unified quality control was carried out on the in situ data. First set a fixed threshold to eliminate the overflow value. Secondly, we tested the time series of site data and eliminated abnormal and missing data due to instrument

damage or bad weather. Finally, we checked the temporal and spatial consistency of the measurement data, delete the meteorological stations with location migration during the study period, and keep the temperature data of meteorological stations with long monitoring time and stable temperature values.

## 3.3 Supplementary data

China's daily near-surface temperature grid dataset was released by the CMA, with a spatial resolution of 0.5°. It is a grid dataset made for the daily maximum, minimum, and average temperatures in China (http://www.nmic.cn/site/index.html, last access: 11 April 2021). The CMA dataset was obtained by combining the daily temperature data monitored by meteorological stations and the digital elevation model (DEM) data generated by re-sampling with three-

dimensional geospatial information through a thin-plate spline interpolation algorithm. The spatial resolution of the CMA data was 0.5°, which is used to make cross-validation.

Moderate Resolution Imaging Spectroradiometer (MODIS) is an important sensor in the Earth Observation System program, which is a medium-resolution imaging spectrometer mounted on the Terra and Aqua satellites. Terra is a morning orbiting satellite that passes through the equator

at approximately 10:30 local time from north to south, and Aqua is an afternoon orbiting satellite that passes through the equator at approximately 1:30 local time from south to north. The Terra satellite has been in service since 1999, and the Aqua satellite has been in service since 2002. Since 2002, the surface temperature data can be obtained 4 times a day from MODIS data through

inversion calculation. In this study, we selected the MOD11A1 and MYD11A1 products, which

can provide daily surface temperature data on a global scale with a spatial resolution of 1 km. To determine the locations of low-quality and missing values in pixels that are affected by cloud pollution and aerosols, MODIS provides quality control fields for each of its products, and quality control documents are mostly encoded in the binary form. MODIS data can be downloaded from the LAADS DAAC website (https://ladsweb.modaps.eosdis.nasa.gov/search/order, last access: 1

December 2020).

In addition to the above data, DEM data were used in this study. The Shuttle Radar Topography Mission (SRTM) DEM used in this study was a radar topographic mapping project jointly implemented by NASA and the National Imagery and Mapping Agency, which was implemented by the Space Shuttle Endeavour. The temperature data were regulated via topographical correction

of SRTM DEM of 90-m resolution to eliminate the influence of topographical fluctuations on air temperature. SRTM DEM data can be obtained through the USGS network (http://www.gscloud.cn/search, last access: 10 February 2021).

## 4.  Methodology

In currently, the $T_{max}$, $T_{min}$, and $T_{avg}$ data can be provided by meteorological stations, other non-

station locations or grid values were estimated by interpolation or indirect methods such as remote sensing. Owing to the limited number of meteorological stations and uneven distribution, it is difficult to guarantee the accuracy of $T_{max}$, $T_{min}$, and $T_{avg}$ obtained through interpolation in some areas. Under rainfall and cloud cover weather conditions, it is impossible to estimate the air temperature from remotely sensed surface temperature data. Even in clear sky conditions, the

formula for estimating near-surface air temperature is not universally applicable, which hinders the accurate development of the $T_a$ dataset to a certain extent. Therefore, to obtain a $T_a$ dataset with a high temporal and spatial resolution and long time series, it is necessary to build a reliable and robust $T_a$ model to estimate $T_{max}$ and $T_{min}$, and further improve the accuracy of $T_{avg}$. Consequently, the product can be more widely used for climate change and research on extreme

weather events. Daily temperature changes are affected by many factors and are extremely sensitive to fluctuations in various weather phenomena. This study calculates $T_{max}$ and $T_{min}$ by distinguishing different weather conditions. First, the daily weather conditions were divided into

the clear sky and non-clear sky conditions. Second, based on the physical process of daily

temperature changes and combined with existing reanalysis data, in situ data, and remote sensing

data, we construct $T_{max}$ and $T_{min}$ models under clear sky conditions. In non-clear sky weather

conditions, a variety of methods are used to determine $T_{max}$ and $T_{min}$. In order to further improve

the accuracy of the data, a modified model is constructed according to the regional situation. More

details are given in the following sections. The overall process of this study is illustrated in Fig.

2. The construction of the dataset was mainly divided into three steps: (1) The process of daily

weather status determination, (2) the process of establishing $T_a$ models under different weather

conditions, and (3) data correction.

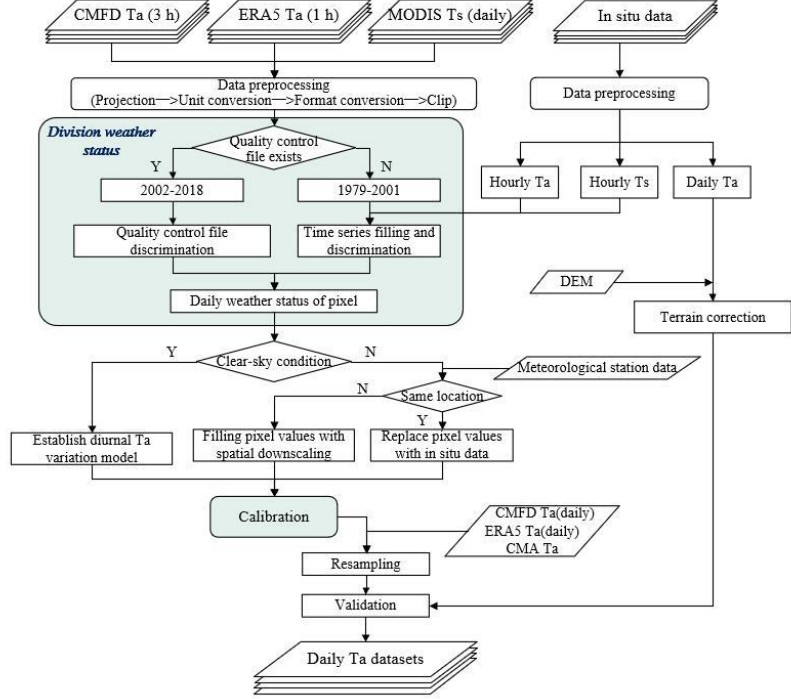

Figure 2. Summary flowchart of $T_a$ dataset establishment.

## 4.1 Strategies for division of weather conditions and $T_a$ estimation

### 4.1.1 Scheme for dividing weather conditions

Different weather conditions have different rules of temperature changes. In order to improve the

estimation accuracy of the maximum and minimum temperature, we conduct specific calculations





by distinguishing daily weather conditions. Clouds and water vapor have a great influence on visible light and thermal infrared remote sensing. Many remote sensing data such as MODIS products generate quality control files for each pixel. Therefore, the quality control field of MODIS can be used to distinguish between clear sky and non-clear sky weather conditions. However, we can only obtain MODIS observation data four times a day since 2002, which cannot cover the time range involved in this study. Therefore, we divided the time series of this study into two periods: 1979–2001 and 2002–2018, and different methods are used for the two-time series to distinguish the daily weather status. For the study period from 2002 to 2018, we distinguished each pixel based on the MODIS quality control field. When the MODIS quality control of all four $T_s$ corresponding to a pixel is in the clear sky condition, the pixel is judged to be in the clear sky condition, otherwise, it is judged to be in the non-clear sky condition.

For the study period from 1979 to 2002, we used the in situ, CMFD, and ERA5 data to determine the daily weather status. First, we filtered each pixel and divided it into two types: meteorological stations corresponding to pixels with and without weather status records. For pixels with weather status records, we used a large number of statistical discrimination methods to analyze the impact of abnormal weather phenomena on temperature fluctuations, which can facilitate the subsequent determination of pixels without weather status records. Statistical analysis shows that there is a significant difference in daily temperature fluctuations between clear sky and non-clear sky conditions, and non-clear weather conditions may cause abnormal temperature fluctuations. Therefore, we converted the judgment of the weather state into the abnormal judgment of the time and frequency of the occurrence $T_{max}$ and $T_{min}$ (The occurrence time of $T_{max}$ and $T_{min}$ is hereinafter cited as $H_{max}$, $H_{min}$). Specifically, when $H_{max}$ and $H_{min}$ occur abnormally or the temperature change is wavy, it is regarded as non-clear sky condition (Zhao and Duan, 2014; Ren et al., 2011). In other cases, they are regarded as clear sky states, and the position of each pixel is marked. Therefore, we needed to further fill the daily time series of each pixel to determine the weather state. In this study, we utilized two strategies to perfect the temperature series obtain the time and frequency of $T_{max}$ and $T_{min}$ for distinguishing the weather conditions. The specific implementation steps for determining weather conditions are shown in Figure 3.




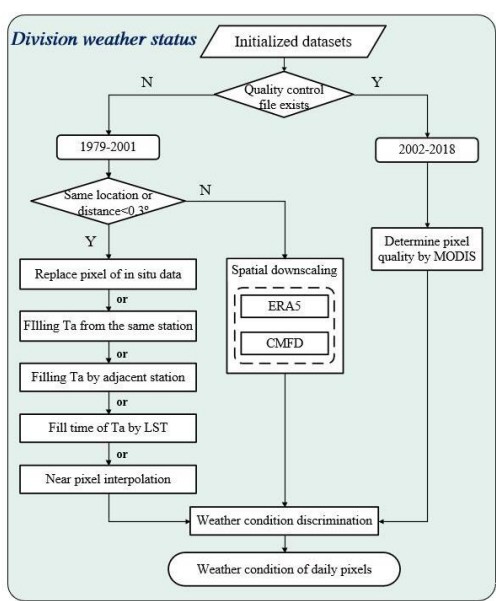

Figure 3. Summary flowchart for the classification of the weather conditions.

In the first strategy, when the pixel location had a corresponding meteorological station or when

the Euclidean distance between adjacent stations was less than 0.3°, we fill in the gaps to improve

the integrity and continuity of the time series. The time series filling process was as follows: (1)

When there were missing values in the measured data at the site, there were no continuous missing

values. In the case of the same spatial range, we use the average of the two times temperatures

before and after the same site to fill in the missing values. (2) When the observation data of a site

were missing continuously, in the case of the same time range, we filled it according to the time

and frequency of the $T_{max}$ and $T_{min}$ occurrence of adjacent sites. This method is mainly based on

the principle that the closer the distance between stations, the stronger the spatial consistency and

correlation of temperature changes. (3) When the station data were continuously missing and the

adjacent station data could not be filled, other relevant data were used for repair within the same

time and space. In this study, we estimated the weather state based on the time and frequency of

the $T_{max}$ and $T_{min}$ from the $T_s$ monitored by the same station. This method theoretically originates

from the approximate consistency between the daily variation ranges of $T_s$ and $T_a$, and is suitable

for situations where there are a large number of missing values and incomplete time series at

meteorological stations and adjacent meteorological stations. Many studies have analyzed the

correlation between the daily trend of $T_a$ and $T_s$ and found that they have strong consistency. The $T_s$ retrieved by remote sensing satellites is also widely used to estimate $T_a$, which proves the reliability of determining the pixel weather state through the $T_s$ time series (He et al., 2020; Yoo et al., 2018; Johnson and Fitzpatrick, 1977; Caesar et al., 2006; Mostovoy et al., 2006). (4) When there is no meteorological station at the pixel location and the distance from the meteorological

station is less than 0.3°, we use the inverse distance weighting method to perform spatial interpolation on adjacent pixels. Determine the weather state by obtaining the time and frequency of each pixel's daily appearance of $T_{max}$ and $T_{min}$.

The second strategy was to target areas where the distribution of sites was sparse and the Euclidean distance between two adjacent sites was greater than 0.3°. In order to make up for the

insufficient coverage and uneven distribution of stations in these areas, this study uses hourly data from ERA5 to refine the time series of each pixel and distinguish the weather status. As there was a certain difference between the spatial resolution of ERA5 and this dataset, it was difficult to meet our demand for higher spatial resolution. Consequently, we developed an effective downscaling process based on the spatial correlation between ERA5 data and CMFD 3-h

temperature data. The ERA5 data (with a spatial resolution of 0.3°) were spatially downscaled with the aid of CMFD data (with a spatial resolution of 0.1°). The downscaling process is illustrated in Fig. 4. First, quality control of the ERA5 and CMFD datasets was performed to eliminate temperature outliers. Second, ERA5 and CMFD data were matched according to time series and central latitude and longitude to construct pixel pairs. Subsequently, we weighted the

high-resolution data to the low-resolution ERA5 data pixel by pixel. Finally, the weight was used to downscale the ERA5 data to the same spatial resolution of the CMFD. The ERA5 downscaling was computed using Eqs.1 and 2.

$$T_E(x_o,y_o) = \frac{T_C(x_o,y_o)}{\sum_{i=0}^{m}\sum_{j=0}^{n} T_C(x_i,y_j)} * T_E(x_m,y_n) \tag{1}$$

$$T_E(x_o,y_o) = \frac{T_M(x_o,y_o)}{\sum_{i=0}^{m}\sum_{j=0}^{n} T_M(x_i,y_j)} * T_E(x_m,y_n) \tag{2}$$

where $T_E$, $T_C$, $T_M$ represents ERA5, CMFD, MODIS data, respectively. $T_E(x_o,y_o)$ is the temperature data after downscaling. $T_E(x_m,y_n)$ is the temperature data before downscaling. i, j are pixel coordinates. m, n are the pixel coordinates before downscaling.

are pixel coordinates. m, n are the pixel coordinates before downscaling.

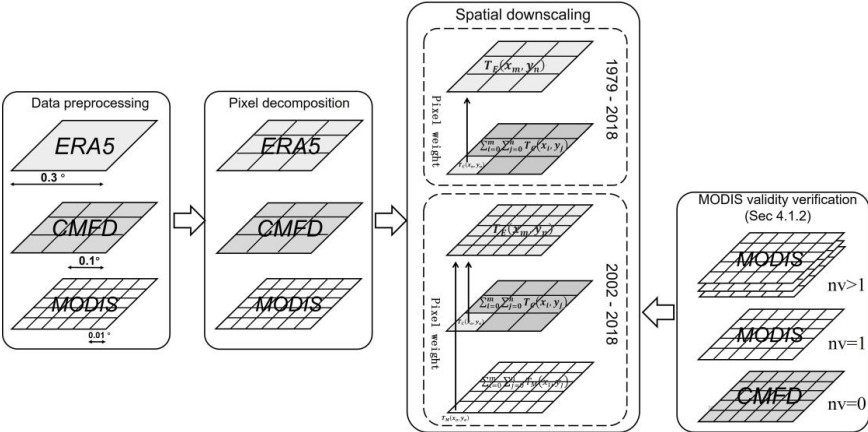

Figure 4. Flowchart for spatial downscaling, where nv represents the number of valid values.

### 4.1.2 $T_{max}$ and $T_{min}$ estimation under clear sky conditions

In addition to the temperature severe fluctuations caused by abnormal weather phenomena, the

daily temperature changes under clear sky conditions have a certain regularity, periodicity, and

asymmetry (Leuning et al., 1995; Johnson and Fitzpatrick, 1977). According to the similarity

between the surface temperature and the diurnal variation trend of air temperature, a method of

estimating $T_a$ is established by the daily air temperature variation model. Verified by

meteorological station data, this method is feasible (Du et al., 2020; Zhu et al., 2013; Perkins et

al., 2007; Cesaraccio et al., 2001; Serrano-Notivoli et al., 2019). However, it is very complicated

to use the surface temperature retrieved by remote sensing methods to estimate the changing trend

of air temperature, and more parameters need to be input, and the relationship between $T_s$ and $T_a$

is not fixed. Therefore, it is difficult to unify the types and quantities of parameters, and it is

difficult to ensure accuracy. As a result, we established a piecewise local sine function of

temperature under clear sky conditions, which can simulate the change in $T_a$ and calculate $T_{max}$

and $T_{min}$ (Mao et al., 2016; Jiang et al., 2010). First, according to the approximate periodicity of

daily temperature changes and the asymmetry of $H_{max}$ and $H_{min}$, we derive the $T_a$ piecewise sine

function of the adjacent regions of $H_{max}$ and $H_{min}$, as shown in Eqs. 3 and 4. Among them, Eq. 3

is the $T_{max}$ function and Eq. 4 is the $T_{min}$ function. Secondly, it is similar to the method of filling

the temperature time series when judging the weather state. By combining in-situ data and

reanalysis data, the temperature sequence is improved and the $H_{max}$ and $H_{min}$ of each pixel are

obtained. These $H_{max}$ and $H_{min}$ values are entered as parameters into the piecewise sine function.

The CMFD 3-h data is used as $T_a$ data, and each pixel $H_{max}$ and $H_{min}$ are used as time, and input

into the piecewise sine function by the least square method for parameterization. We can obtain

the values of $A_t$ and $B_t$ used to construct the piecewise sine function. The least squares method is

a mathematical optimization technique, which uses the least square sum of residuals as the

estimation standard for the best matching function. It is usually used in statistical models and is

by far the most applicable and widely used parameter estimation method (Qiu and Jiang, 2021;

Ge, 2015; Floyd and Braddock, 1984). Finally, $H_{max}$ and $H_{min}$ values were substituted into the

derivation formula to obtain $T_{max}$ and $T_{min}$ as preliminary results for subsequent correction and

analysis. By constructing a temperature model pixel by pixel to meet the temporal and spatial

heterogeneity of each region.

$$T_{max} = A_t * \sin[\frac{(H_o - H_{max})\pi}{H_{max} - H_{min}} - \frac{\pi}{2}] + B_t \qquad (3)$$

$$T_{min} = A_t * \sin[\frac{(H_o - H_{max})\pi}{24 - H_{max} + H_{min}} - \frac{\pi}{2}] + B_t \qquad (4)$$

where $H_{max}$ is the occurrence time of the daily maximum temperature. $H_{min}$ is the occurrence time

of the daily minimum temperature. $H_o$ is the input time, and $A_t$ and $B_t$ are unknown parameters.

**4.1.3 $T_{max}$ and $T_{min}$ estimation under cloudy-sky conditions**

The daily temperature fluctuations in non-clear-sky conditions are relatively large, and there may

be large-scale cooling or sudden temperature changes in a short period of time. Based on the

spatial location information of each pixel, in situ data are the most reliable and representative data

source. Therefore, if there are in situ data at the pixel location, the temperature data at the same

time will be directly obtained from the site to replace the pixel values $T_{max}$ and $T_{min}$. For the pixels

corresponding to non-meteorological stations, similar to the method of spatial downscaling for

the pixel positions of non-meteorological stations in the weather status judgment, we use ERA5

data to perform spatial downscaling with the assistance of CMFD data. By adding high spatial

resolution MODIS data, the downscaling method is further expanded to improve the accuracy of

each pixel. However, for the method of using remote sensing data to assist downscaling, we

needed to consider the degree of influence of cloudy-sky weather phenomena. First, we performed

effective value statistics on the MODIS data. When not all pixels of the MODIS data were valid,

the pixels with poor-quality or missing data were identified and removed. The corresponding time

of the effective pixel was matched with the ERA5 data according to the nearby time to obtain the

data weight for spatial downscaling. When the pixels in MODIS were invalid in 1 day, we used

CMFD data for downscaling and finally obtained $T_{max}$ and $T_{min}$. The downscaling process and the

validity determination of MODIS data are shown in    Figure 4, and the downscaling formulas are

shown in Eqs. 1 and 2.

### 4.1.4 $T_{avg}$ estimation

Usually, the calculation of average temperature is to use the temperature value observed every

day to do an arithmetic average. If each pixel has hourly temperature data, the calculated daily

average temperature is the most representative. Because it is difficult to obtain hourly data, people

often use 4-hours temperature or directly use the maximum and minimum average values as the

daily average temperature. In order to improve the accuracy of the average temperature as much

as possible, we use the three-hour temperature data provided by CMFD and the maximum and

minimum values calculated above to do an arithmetic average to get the daily average temperature.

Finally, multiple linear regression correction was performed on the $T_{avg}$ output value according to

the in situ data to improve the accuracy (the linear correction method was the same as that

described in Sect. 0), and the daily $T_{avg}$ dataset was obtained.

### 410 4.2 $T_a$ data calibration scheme

Surface temperature is sensitive to changes in altitude and is easily affected by the surrounding

environment. For non-meteorological station pixels, we use interpolation to fill in the pixel values

based on the principle of regional consistency. In order to improve the accuracy of pixel

temperature at non-meteorological sites, we fully consider the influence of altitude on temperature.

First, the in-situ $T_a$ is unified to sea level according to the vertical rate of temperature drop. Then,

the non-site pixels are interpolated according to the station data, and finally, the interpolated pixel

values are restored to the corresponding elevation. This method can reduce the influence of

altitude on temperature to a certain extent and improve the accuracy of the dataset. In this study,

we used a uniform vertical temperature drop rate ($\gamma$), that is, for every 100 m increase in altitude,

the atmospheric temperature drops vertically by 0.65°C, and vice versa. The height correction





formula is given by Eq. 5 (He and Wang, 2020; Schicker et al., 2015; Wang et al., 2013).

$$T_{SL} = T_a - \gamma * \left( H_{SL} - H_a \right) \tag{5}$$

where $T_{SL}$ is the sea level temperature, $T_a$ is the temperature of the meteorological station, and $H_{SL}$ is the sea level height, where the value of $\gamma$ is approximately 0.0065°C/m.

Based on the jackknife method, 699 in situ stations across the country were divided into 140
verification points and 559 calibration points according to the ratio of in 20% and 80% to establish a multiple linear regression equation (Benali et al., 2012; Xu et al., 2017). From the preliminary accuracy results of the temperature change model in Sect. 0, it can be seen that although the overall accuracy was high, there is still the problem of abnormal temperature values of the model output data caused by the violent fluctuations in daily temperature changes. Further correction is required
to reduce the deviation and improve the accuracy of the dataset. The data correction process is illustrated in Figure 5. For the abnormal temperature value, we replace the $T_a$ at the pixel location with the observation $T_a$ from the meteorological station, and performed the adjacent pixel temperature correction for the pixel without the meteorological station at the pixel location. The multiple linear regression method is used to perform multiple linear regression on the original
temperature, and the stepwise regression relationship between the measured value of the station and the fitted value of the corresponding pixel is established. Then calculate the predicted value of the regression temperature according to the regression equation, and obtain the temperature residual value by calculating the observed value and the predicted value. The residual value and the predicted value are spatially added to obtain the final corrected temperature (Cristobal et al.,
2006). The modified expression is shown in Eq. 6.

$$V(x, y) = \hat{m}(x, y) + \hat{\varepsilon}(x, y) \tag{6}$$

where x and y are the numbers of rows and columns of pixels, respectively, $V(x, y)$ is the correction value of the regression equation, $\hat{m}(x, y)$ is the regression prediction value of air temperature, and $\hat{\varepsilon}(x, y)$ is the residual value.

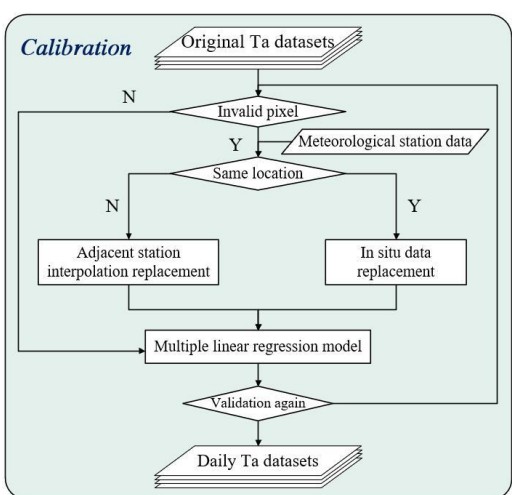

Figure 5. Flowchart for calibration of $T_a$ model data.

## 4.3 Evaluation metrics

To verify the accuracy of this dataset, we first verified the accuracy of the original temperature dataset and the corrected dataset in this study with the in situ data. A scatter diagram was used to compare the results before and after validation. The scatter diagram represents the overall

distribution and aggregation of the data and can intuitively convey accurate information of the data. Further, to better evaluate the accuracy of this dataset, we selected areas with uniform surface types and flat terrain under clear skies as the comparative study area and compared this product with the existing datasets. We selected three indicators as metrics to measure the accuracy of variables: $R^2$, MAE, RMSE.

We compared $T_{max}$ and $T_{min}$ with ERA5 data and CMA data. It is worth noting that the ERA5 reanalysis dataset is an hourly temperature grid dataset, so we obtain the highest and lowest temperature values of ERA5 by constructing a local sine function similar to the previous section, and further calculate the average daily temperature. The accuracy of $T_{avg}$ products in this study is verified with ERA5 data, CMA data, and CMFD daily temperature data. Since the spatial

resolution of CMA is 0.5°, in order to facilitate comparison, we resample the spatial resolution of all datasets to 0.5°.





## 4.4 Analysis of the $T_a$ series trend

We not only compared the output $T_a$ data with the in situ data, but also assessed the climate change trends of $T_{max}$, $T_{min}$, and $T_{avg}$ in various regions of China, and further tested the effectiveness and regional applicability of the dataset through various climate variables. This study used four temperature indexes (TXx, TNn, TX90p, and TN10p) to analyze the trends of $T_{max}$ and $T_{min}$ extreme temperature changes each year. Specifically, TXx (TNn) abnormality refers to the difference between the sum of monthly $T_{max}$ ($T_{min}$) and the multi-year average of monthly $T_{max}$ ($T_{min}$) in each year. The multi-year period of this study is 40 years. In addition, linear regression was performed on the TXx (TNn) anomaly to analyze the inter-annual variation trend. The TX90p (TN10p) arranged the daily $T_{max}$ ($T_{min}$) of each month during the study period in ascending order of temperature, and we selected the portions with more than 90% (less than 10%) correlation with the number of days in each year.

To study the spatiotemporal variation trend of $T_{avg}$, we used linear regression analysis (K), correlation coefficient analysis (R), and T-test (Du et al., 2020; Yan et al., 2020; Cao et al., 2021). The interannual change rate and correlation of $T_{avg}$ were calculated by K and R, and the formula is given by Eqs. 7 and 8, respectively. We performed a two-tailed significance test on the T-test to quantify the significance of the temperature and time-series changes (Eq. 9).

$$K = \frac{n\sum_{i=1}^{n}(iT_i) - \sum_{i=1}^{n}i\sum_{i=1}^{n}T_i}{n\sum_{i=1}^{n}i^2 - (\sum_{i=1}^{n}i)^2} \qquad (7)$$

$$R = \frac{n\sum_{i=1}^{n}(iT_i) - \sum_{i=1}^{n}i\sum_{i=1}^{n}T_i}{\sqrt{n\sum_{i=1}^{n}i^2 - (\sum_{i=1}^{n}i)^2} * \sqrt{n\sum_{i=1}^{n}T_i^2 - (\sum_{i=1}^{n}T_i)^2}} \qquad (8)$$

$$T\_test(R) = \frac{R\sqrt{n-2}}{\sqrt{1-R^2}} \qquad (9)$$

where n represents the total number of years of the time series length, i represents the year, and $T_i$ represents $T_{avg}$ in the i-th year. K > 0 indicates that the temperature is increases within the time series, and K < 0 indicates that the temperature is decreases within the time series.

## 5. Results

### 5.1 Accuracy verification before calibration

According to the six subregions divided in Fig. 1, comparative analysis of this product ($T_{max}$, $T_{min}$

and $T_{avg}$) based on in-situ data are made respectively. Fig. 6 shows the accuracy scatter plot

between the original data of $T_{max}$ and the in situ data. The $R^2$ fluctuated from 0.91 to 0.99, the

MAE ranged from 1.69 °C to 2.71 °C, and the RMSE ranged from 2.15 °C to 3.20 °C. Fig. 7

shows the accuracy scatter plot of $T_{min}$. The $R^2$ fluctuated from 0.93 to 0.97, the MAE ranged

from 1.34 °C to 2.17 °C, and the RMSE fluctuated from 1.68 °C to 2.79 °C. Fig. 8 shows the

accuracy scatter plot of $T_{avg}$. The $R^2$ fluctuated between 0.97 and 0.99, the MAE ranged from

0.58 °C to 0.96 °C, and the RMSE fluctuated from 0.86 °C to 1.60 °C. It can be seen from Figs.

6, 7, and 8 that the $R^2$ of $T_{max}$, $T_{min}$, and $T_{avg}$ and the temperature measured at the meteorological

station were all greater than 0.90. In general, our method performed well in estimating the daily

temperature values. However, due to the impact of complex changes in weather, the distribution

of temperature values on certain days is more discrete, especially in the study areas V and VI.

Further corrections are needed to reduce errors and improve the accuracy of the dataset.

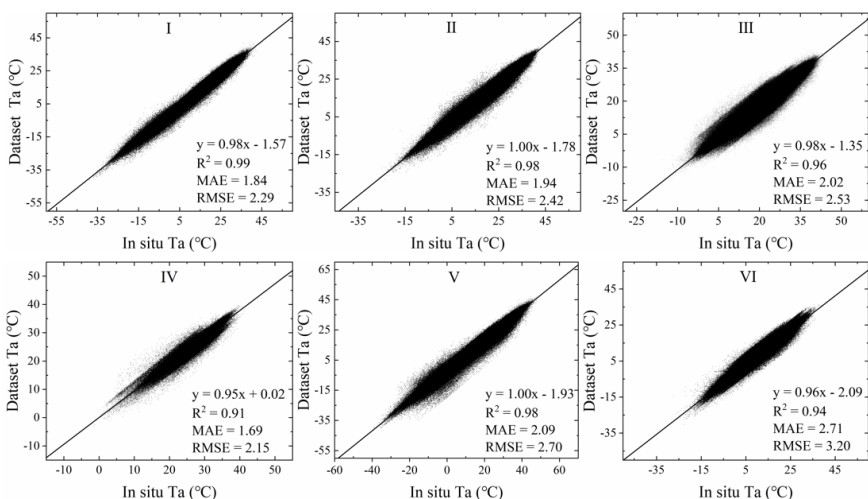

Figure 6. Scatter diagrams of the $T_{max}$ output from the $T_a$ model against ground station data; the statistical accuracy

measures ($R^2$, MAE, and RMSE) are also indicated.

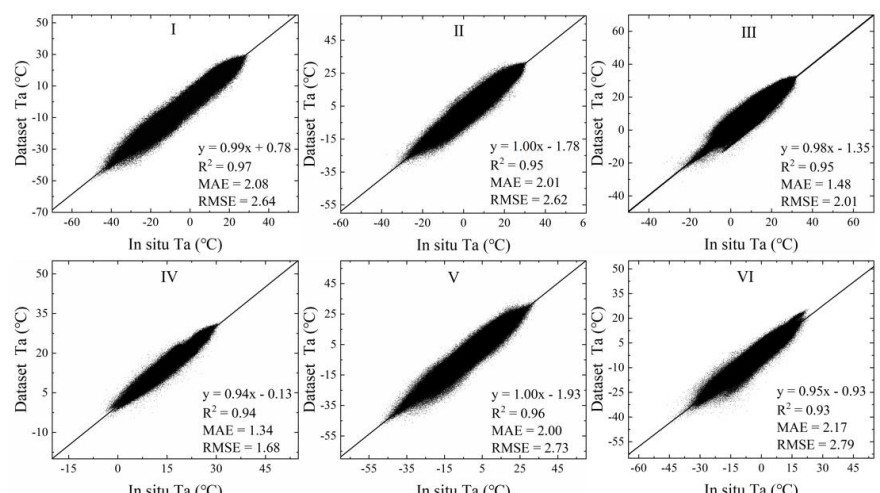

Figure 7. Scatter diagrams of the $T_{min}$ output from the $T_a$ model against ground station data; the statistical accuracy

measures ($R^2$, MAE, and RMSE) are also indicated.

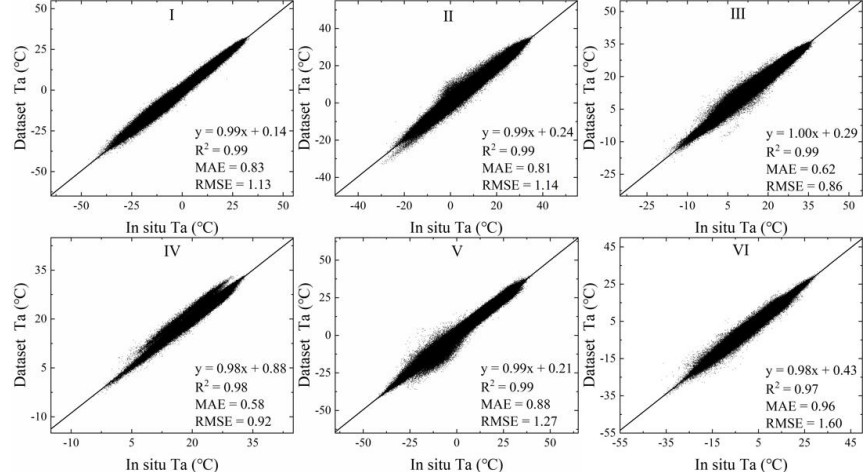

Figure 8. Scatter diagrams of the $T_{avg}$ output from the $T_a$ model against ground station data; the statistical accuracy

measures ($R^2$, MAE, and RMSE) are also indicated.

## 5.2 Accuracy verification after calibration

The temperature was further corrected using the linear correction method. The data verification

results of $T_a$ after correction are shown in Figs. 9, 10, and 11. The results showed that the corrected

data had a higher consistency with the in situ data. The fitted and observed temperatures were

linearly distributed and gradually approached the regression line, and the outliers were greatly





reduced. Fig. 9 shows the corrected scatter plot of $T_{max}$ for each study area. The $R^2$ fluctuated from 0.96 to 0.99, and the MAE ranged from 0.63 °C to 1.40 °C, the RMSE fluctuated from 0.86 °C to 1.78 °C. Fig. 10 shows the corrected scatter plot of $T_{min}$ for each study area. The $R^2$ fluctuated between 0.95 and 0.99, and the MAE ranged from 0.58 °C to 1.61 °C, the RMSE

fluctuated from 0.78 °C to 2.09 °C. Fig. 11 depicts the corrected scatter plot of $T_{avg}$ in each study area, where $R^2$ fluctuated between 0.99 and 1.00, the MAE ranged from 0.27 °C to 0.68 °C, the RMSE fluctuated from 0.35 °C to 1.00 °C. The results showed that the distribution of numerical points in each area after the correction was denser, mostly concentrated near the 1:1 line, and the degree of clustering with the measured data was higher than before calibration. When we

performed a detailed analysis of the daily temperature in the six study areas, we found that the accuracy measurement values differed greatly between the east and west. For example, the accuracy error of study area IV is small, and the accuracy error of study area VI and V is large, which may be affected by the regional topography and the distribution of meteorological stations. The IV study area is located in the tropical monsoon climate zone, affected by latitude and

topography, the temperature is relatively high throughout the year. Moreover, the area is located in the eastern part of China with densely distributed meteorological stations and relatively flat terrain. Linear correction can significantly improve the agreement between the estimated value and the observed value. The study areas VI and V have higher RMSE. They are located in the Qinghai-Tibet Plateau in southwest China and Xinjiang in the northwest. Such areas have similar

characteristics, such as high altitude, large spatial heterogeneity, and few meteorological stations. It shows that the temperature has strong spatial heterogeneity. In general, the corrected dataset has higher accuracy, satisfies the spatial heterogeneity of different regions, and better estimates the temperature under different weather conditions.

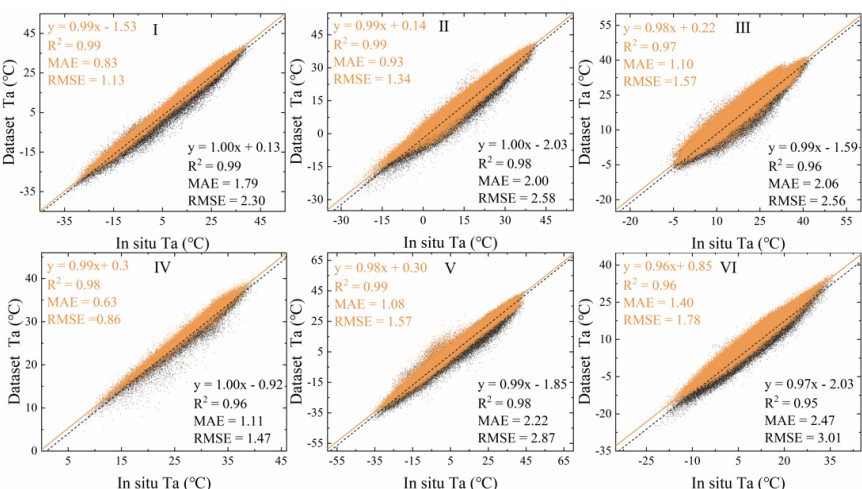

Figure 9. Scatter diagrams of the original $T_{max}$ and reconstructed results versus their corresponding ground station data in six natural subregions (I, II, III, IV, V, and VI). The gray points indicate low-quality pixel values in the original $T_{max}$ data, and the orange points represent the values in the after-calibrated $T_{max}$ dataset, and the statistical accuracy measures ($R^2$, MAE, and RMSE) are also indicated.

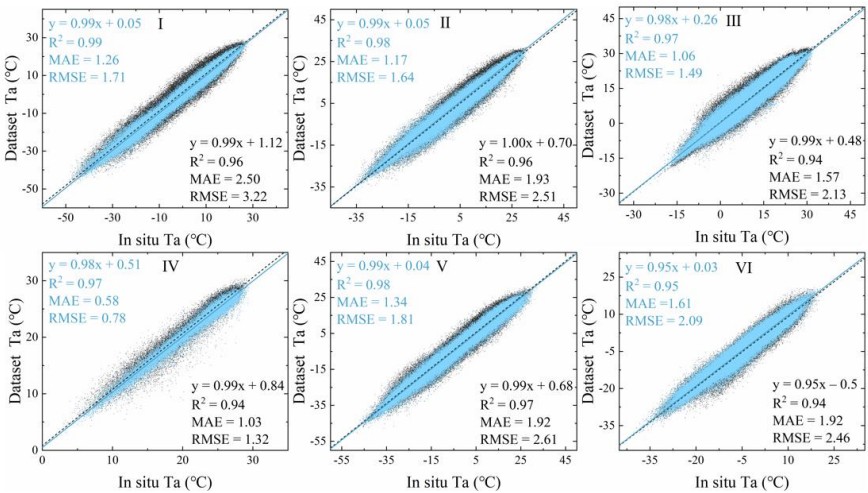

Figure 10. Scatter diagrams of the original $T_{min}$ and reconstructed results versus their corresponding ground station data in six natural subregions (I, II, III, IV, V, and VI). The gray points indicate low-quality pixel values in the original $T_{min}$ data, and the blue points represent the values in the after-calibrated $T_{min}$ dataset, and the statistical accuracy measures ($R^2$, MAE, and RMSE) are also indicated.

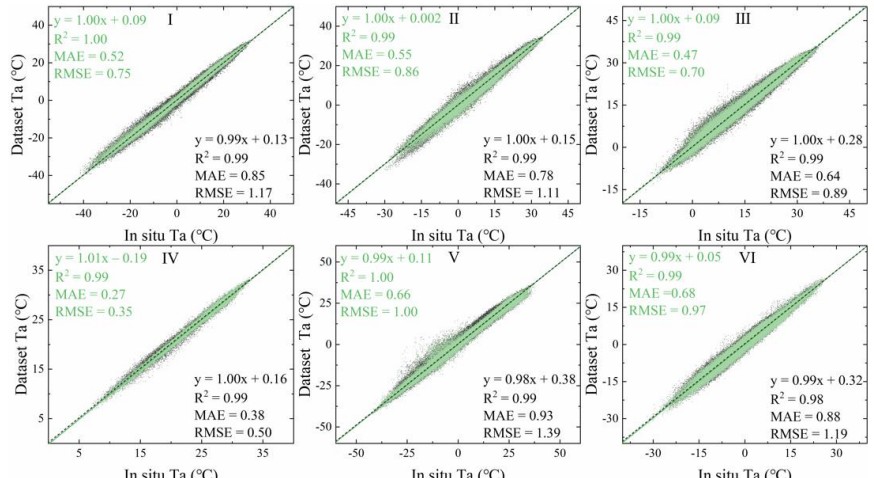

Figure 11. Scatter diagrams of the original $T_{avg}$ and reconstructed results versus their corresponding ground station data in six natural subregions (I, II, III, IV, V, and VI). The gray points indicate low-quality pixel values in the original $T_{avg}$ data, and the green points represent the values in the after-calibrated $T_{avg}$ dataset, and the statistical accuracy measures ($R^2$, MAE, and RMSE) are also indicated.

To further verify the robustness and accuracy of this product, Table 1 shows the cross-validation results of this product and other datasets, and the mean average precision (MAP) of each region. It can be seen from the table that this product has a high regional consistency with other datasets. Study area IV located in the tropical monsoon climate zone has higher accuracy, while study area VI located in the Qinghai-Tibet Plateau region of China has lower data accuracy. This may be because the reanalysis dataset is also affected by the number and distribution of meteorological stations, and the spatial heterogeneity. The accuracy and robustness of the product has been confirmed from another angle. The accuracy comparison of each area shows that this product has higher accuracy and spatial representation than other datasets. $R^2$ is closer to 1, and both MAE and RMSE remain low. Through the accuracy evaluation and data comparison between this product and the existing dataset, it is found that our product has a better temperature estimation of each area, and the overall accuracy and accuracy of the dataset is higher.

Table 1. Cross-validation results of this product and other datasets.

| Temp. Type | Index | Data | I | II | III | IV | V | VI | MAP |
|---|---|---|---|---|---|---|---|---|---|
| | | ERA5 | 0.99 | 0.97 | 0.94 | 0.94 | 0.97 | 0.94 | 0.96 |


| | | | | | | | | | |
|---|---|---|---|---|---|---|---|---|---|
| MAX | $R^2$ | CMA | 1.00 | 0.95 | 0.95 | 0.98 | 0.99 | 0.90 | 0.96 |
| | | DATASET | 0.99 | 0.99 | 0.97 | 0.98 | 0.99 | 0.96 | 0.98 |
| | MAE | ERA5 | 1.05 | 1.25 | 1.47 | 0.99 | 1.53 | 1.99 | 1.38 |
| | | CMA | 0.67 | 1.28 | 1.28 | 0.63 | 0.81 | 1.58 | 1.04 |
| | | DATASET | 0.73 | 0.94 | 1.07 | 0.62 | 1.02 | 1.40 | 0.96 |
| | RMSE | ERA5 | 1.69 | 1.52 | 2.14 | 1.68 | 1.91 | 2.30 | 1.87 |
| | | CMA | 0.99 | 1.80 | 1.76 | 0.83 | 1.22 | 2.79 | 1.57 |
| | | DATASET | 1.03 | 1.14 | 1.37 | 0.81 | 1.57 | 1.78 | 1.28 |
| MIN | $R^2$ | ERA5 | 0.96 | 0.95 | 0.96 | 0.95 | 0.97 | 0.90 | 0.95 |
| | | CMA | 0.99 | 0.97 | 0.96 | 0.98 | 0.99 | 0.90 | 0.97 |
| | | DATASET | 0.99 | 0.98 | 0.97 | 0.97 | 0.98 | 0.95 | 0.97 |
| | MAE | ERA5 | 1.68 | 1.28 | 1.48 | 1.00 | 1.48 | 2.09 | 1.50 |
| | | CMA | 0.85 | 1.24 | 1.18 | 0.46 | 0.98 | 2.23 | 1.16 |
| | | DATASET | 1.13 | 1.14 | 1.04 | 0.57 | 1.34 | 1.41 | 1.10 |
| | RMSE | ERA5 | 1.95 | 1.98 | 1.73 | 1.32 | 2.21 | 2.34 | 1.92 |
| | | CMA | 1.19 | 1.99 | 1.72 | 0.63 | 1.47 | 2.80 | 1.63 |
| | | DATASET | 1.31 | 1.60 | 1.49 | 0.74 | 1.61 | 2.05 | 1.47 |
| AVG | $R^2$ | CMFD | 0.99 | 0.99 | 0.98 | 0.99 | 0.97 | 0.98 | 0.98 |
| | | ERA5 | 0.98 | 0.97 | 0.97 | 0.99 | 0.97 | 0.97 | 0.98 |
| | | CMA | 1.00 | 0.97 | 0.96 | 0.99 | 0.99 | 0.91 | 0.97 |
| | | DATASET | 0.99 | 0.99 | 0.98 | 0.99 | 0.98 | 0.98 | 0.99 |
| | MAE | CMFD | 0.46 | 0.49 | 0.44 | 0.30 | 0.53 | 0.89 | 0.52 |
| | | ERA5 | 0.50 | 0.52 | 0.48 | 0.45 | 0.70 | 0.73 | 0.56 |
| | | CMA | 0.59 | 1.07 | 1.09 | 0.41 | 0.79 | 1.34 | 0.88 |
| | | DATASET | 0.51 | 0.56 | 0.53 | 0.27 | 0.65 | 0.67 | 0.53 |
| | RMSE | CMFD | 0.60 | 1.19 | 0.75 | 0.41 | 1.26 | 1.17 | 0.90 |
| | | ERA5 | 0.57 | 1.17 | 0.71 | 0.52 | 1.24 | 1.15 | 0.89 |
| | | CMA | 0.88 | 1.30 | 1.30 | 0.54 | 1.23 | 1.64 | 1.15 |
| | | DATASET | 0.65 | 0.79 | 0.70 | 0.35 | 1.20 | 1.06 | 0.79 |

## 5.3 Application of the product for trend analysis

We analyze temperature changes in various regions of China through extreme climate indexes and change trend values to further test the validity and regional applicability of the dataset, as shown in Figs. 12 and 13, which show that the TXx anomalies and TNn anomalies are consistent in the regional change trend. Although the annual anomalies fluctuated during the study period, they gradually changed from negative to positive. This confirmed that the temperature fluctuated and increased, and the $T_{max}$ and $T_{min}$ gradually increased, which is consistent with the global warming trend. The average temperature rise of TXx anomalies in each study area was 0.42°C/a, and the



average temperature rise of TXx anomalies was 0.47°C/a. The histograms in Figs. 12 and 13 show

that the number of warm days and cold nights fluctuates increasing and decreasing trend,

respectively. In addition, there are similarities in the change trends between warm days and cold

nights. For example, in 1980, under the continuous influence of strong cold air in the north, low-

temperature weather occurred continuously in most areas of China, and many areas experienced

low-temperature disasters, which leads to a decrease in the number of warm days and an increase

in the number of cold nights. In 2015, 2016, and 2017, the temperature continued to rise, with

high temperatures that occur once in decades. This is closely related to the severe El Niño events

that occurred in 2015 and 2016, the impact of the subtropical high in 2017, and the overall global

warming trend. At the same time, there has been an increase in the number of warm days and a

decrease in the number of cold nights. Meteorological events can indirectly verify the accuracy

of this product, indicating that the corrected data can be used to analyze long-term temporal and

spatial changes in temperature.

In order to further analyze the change rate and regional differences of $T_{avg}$ during the study

period, we conducted an analysis of the temperature change rate (K), correlation coefficient (R),

and significance test of the correlation coefficient (T-test(R)). As shown in Figure 14 (a) and (a'),

$T_{avg}$ in most regions of China showed a weak positive warming trend, accounting for 92.13% of

the total, and the average temperature of $T_{avg}$ in each region was rising by 0.03°C/a. Through the

analysis of the R in Figure 14 (b) and (b'), it is observed that they show a strong correlation in the

area of 48.77% and a correlation in the area of 84.06%, which shows that there is a high correlation

between temperature changes and time. Figure 14 (c) and (c') show that after performing a

significance test on the R between temperature and time, 83.17% of the area passed the 95%

significance test, and 75.23% of the area passed the 99% significance test, which shows that the

correlation between temperature and time development is significant.

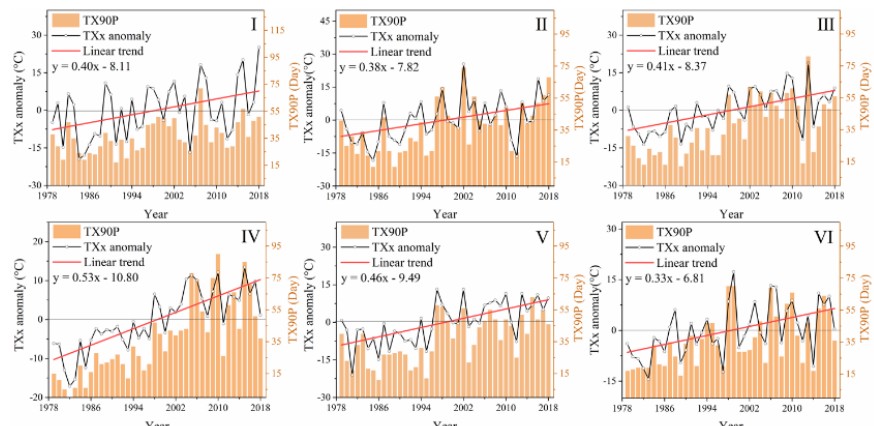

Figure 12. Multi-axis diagram of TXx anomaly, TX90p, and T$_{max}$ linear trend graphs. The broken black line represents TXx anomaly, the red line represents the linear regression of the TXx anomaly, and the orange histogram represents the TX90p change trend.

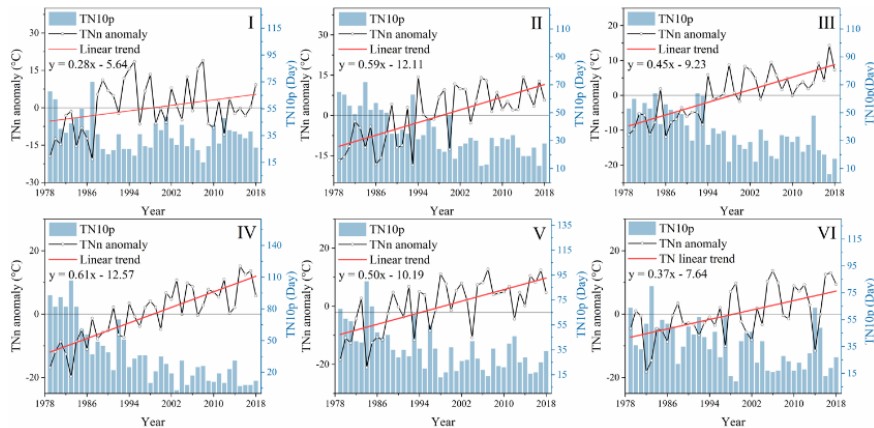

Figure 13. Multi-axis diagram of TNn anomaly, TN10p, and T$_{min}$ linear trend graphs. The broken black line

represents TNn anomaly, the red line represents the linear regression of the TNn anomaly, and the blue histogram represents the TN10p change trend.

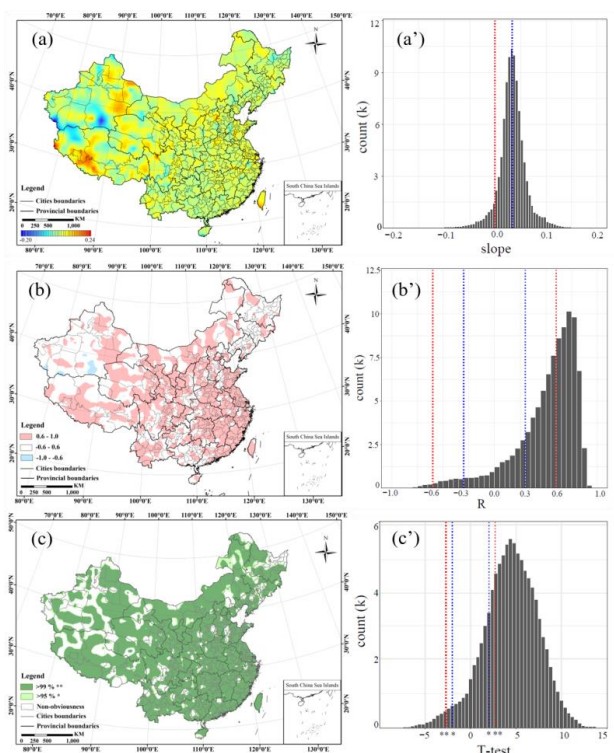

Figure 14. Multi-year climate change trends in $T_{avg}$. (a) K, calculated by Eq. 7; (b) R between temperature change and time series development, calculated by Eq. 8; (c) T-test (R), calculated by Eq. 9. (a'), (b') and (c') respectively represent the distribution of pixel values in the corresponding (a), (b) and (c) spatial images.

## 6. Data availability

The daily $T_a$ products at 0.1° resolution from 1979 to 2018 are freely available to the public in the tif format at https://doi.org/10.5281/zenodo.5502275 (Fang et al., 2021), which are distributed under a Creative Commons Attribution 4.0 License.

## 7. Code availability

We are finishing and improving the code. If the paper is accepted, the code will be made public soon.

## 8. Conclusions

$T_a$ is an indispensable variable for global climate change research. Therefore, it is very important for how to obtain high-precision and high-temporal resolution air temperature data products.



Many researchers have made a lot of efforts, and have produced some datasets through different data sources for the global or local region. But with the need for refinement of research, we need to further improve the accuracy and spatio-temporal resolution. Based on the full analysis of the advantages and disadvantages of various datasets and data sources, this study integrates various

data sources, such as in-situ data, remote sensing data, and reanalysis data, and proposes a reconstruction model of $T_a$ under a clear sky and non-clear sky weather conditions, respectively. A multiple linear regression model was used to further improve the accuracy of the data, and we obtained a new set of grid high-resolution daily temperature datasets in China from 1979 to 2018. For $T_{max}$, validation using in situ data shows that the RMSE ranges from 0.86 ℃ to 1.78 ℃, the

MAE varies from 0.63 ℃ to 1.40 ℃ and the $R^2$ ranges from 0.96 to 0.99. For $T_{min}$, RMSE ranges from 0.78 ℃ to 2.09 ℃, the MAE varies from 0.58 ℃ to 1.61 ℃ and the $R^2$ ranges from 0.95 to 0.99. For $T_{avg}$, RMSE ranges from 0.35 ℃ to 1.00 ℃, the MAE varies from 0.27 ℃ to 0.68 ℃ and the $R^2$ ranges from 0.99 to 1.00. Furthermore, we verified the $T_a$ dataset with the existing reanalysis dataset and found that the proposed dataset has credibility and accuracy. Moreover,

based on the particularity of geographic climate change in different regions, we used four extreme climate indicators (TXx and TNn anomalies, TX90p, and TN10p) and three climate change indices (K, R, and T-test) to analyze the trend changes of $T_{max}$, $T_{min}$, and $T_{avg}$, respectively. In summary, the temperature in most regions of China had been gradually increasing. The number of cold nights and warm days gradually decreased and increased, respectively, and the $T_{max}$ and

$T_{min}$ gradually increased, which is consistent with the general trend of global warming.

However, due to various factors, the weather may occasionally change drastically, such as hail. Historical data cannot provide more specific weather information, especially in areas where there are no meteorological stations, it is difficult to refine past data. However, in future research, we need to consider more meteorological satellite data, especially geostationary meteorological

satellites, which will help improve the accuracy of surface temperature datasets.

**Author contributions.** SF and KM designed the research, developed the methodology and wrote the manuscript; and XX, PW, JS, SMB, TX, MC and EH revised the manuscript.

**Competing interests.** The authors declare no conflicts of interest.

**Acknowledgements.** The authors thank the China Meteorological Administration for providing
CMA data and the ground measurements data, the Institute of Tibetan Plateau Research, Chinese Academy of Sciences for providing CMFD dataset, and the NASA Earth Observing System Data and Information System for providing the MODIS LST and DEM data. We also thank the ECMWF for providing the climate reanalysis data.

**Financial support.** This work was supported by the Second Tibetan Plateau Scientific Expedition
and Research Program (STEP)"Dynamic monitoring and simulation of water cycle in Asian water tower area" (grant no. 2019QZKK0206), the National Key Project of China (grant no. 2018YFC1506602), the National Natural Science Foundation of China (grant no. 41921001), the Fundamental Research Funds for Central Nonprofit Scientific Institution (grant no. 1610132020014) and the Open Fund of the State Key Laboratory of Remote Sensing Science (grant no.
OFSLRSS201910).

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
