# Peer review of "Dataset of daily near-surface air temperature in China from 1979 to 2018"

_Earth System Science Data, 2021_

## Community Comment (CC2)

This is a very good work. It is currently the most comprehensive set of air temperature datasets with the longest time and the highest spatial resolution of the daily maximum temperature, daily minimum temperature and average temperature in China. We know that in different time periods, the number of meteorological stations is inconsistent, the number of early observation stations is very limited, and the data records are also inconsistent. The authors sorted out and screened the previous data, especially when constructing the daily maximum and minimum temperature models, dynamically considering changes in weather, which overcomes the shortcomings of traditional methods and compensates for missing data where there are no weather stations.

In order to further improve the accuracy, the author divides China into 6 regions according to climate conditions and topography, and builds different models for different regions to make further corrections. This is a great work, and this dataset will bring great convenience and support to related research in China. At present, the data set has been downloaded more than 16,000 times, which shows the urgent need and recognition of many researchers for this dataset. Thank you very much for your work.

---

## Author Comment (AC3)

Dear referees,

Thank you for your valuable comments on our manuscript. First, we would like to express our sincere appreciation for your professional and insightful remarks on our paper. These comments are all valuable and have helped us to improve the quality of our paper. We have studied each comment and have made revisions that we hope will meet with approval. Please find our detailed responses below. For convenience, we also attach a version of the manuscript with changes incorporated. Thanks again.

With our best regards,
Shu Fang and co-authors
* * *
**Response to referees**

**Point 1**:The idea and practice of applying (1) observation data like satellite remote sensing data, conventional ground observation data, (2) DEM data and (3) reanalysis products to build a new datasets through constructing near-surface air temperature model with a certain physical relationship should be encouraged. The researchers have finished a lot of work. However, part of the method in Figure 2 and Figure 3 are not rigorous for the large coverage and for the long time series so the model could be further studied according to some reasonable physical logics. And, The English scientific term usage level and grammar application in this paper needs to be improved further.

**Response 1: Thank you for your good comments and guidance. The daily maximum and minimum temperatures, as well as their average values, are very important for research on climate change and regional energy balance. There are very few maximum-minimum-average air temperature datasets with high spatiotemporal resolution. In order to provide a complete dataset, we have done this work. This dataset is very popular with more than 43,000 downloads (https://zenodo.org/record/5502275#.YasWCLq-uUk). According to your suggestion, we have tried our best to modify and improve it.**

**Point 2**:Whether the temperature observation data from meteorological stations have been underwent homogeneous data processing and tested for homogeneity except the steps mentioned in line 207-214? It is very important.

**Response 2: Thank you for your guidance. The meteorological observation station data we obtained from the China Meteorological Administration have undergone homogeneity data processing and homogeneity testing. Moreover, we have done further checks and have not written very detail here. we have made supplementary explanations in the manuscript, thank you.**

**Point 3**:It is better to apply cloud mask product from the geostationary satellites to get clear/cloud detection for the complete diurnal variation observation, while the spatial

resolution is limited. The clear/cloud detection from TERRA and AQUA polar satellites could meet the requirement well enough? Especially when: (1) the area is large enough in China in the scope of Figure 1, (2) the spatial resolution of the dataset is 0.1°. The swath of MODIS is only 2350 km, so the spatial gap of missing observation is large, even "same location or Euclidean distance < 0.3°". It is only a compromise to apply the remote sensing data from polar satellite in this research. The coverage is suspicious.

**Response 3: Thank you for your guidance. You have given us a very good suggestion, that is, in theory, geostationary satellites can indeed better perform the clear/cloud detection for all-sky change observations. However, due to some reasons, it is difficult for us to obtain continuous and stable geostationary meteorological satellite data in China for the past 20 years. MODIS data in TERRA and AQUA are stable and continuous, which are currently the best long-term satellite data. Although the swath of MODIS is only 2350 km, the satellites are sun-synchronized and there are very little missing data. In addition, MODIS data products have done strict data quality control, which is more assured to use, and they are free and publicly downloadable. Furthermore, when the data is missing, we still use the assimilation data ERA5 as the control condition. We didn't descript very detail here, we made a supplementary explanation. In short, your suggestion is very good. In the future, the datasets should try to use geostationary satellite data to judge weather conditions. Thanks again.**

**Point 4:** The construction of near-surface air Temperature models in cloudy days or under complex weather conditions, including the determinations of Tmax and Tmin, are relatively difficult. The research object also involves the diurnal variation of T including Tmax and Tmin, but no detection for each location for 4 times a day or less. The single polar satellite flies over each area twice a day – day and night – it is temporal missing. The strategy under clear sky condition in this study is treated relatively to make sense, although some hypothesis may fail the test, like 4 clear sky conditions in the same location in a day, for the temperature peak/valley variation rule differ a lot due to many reasons. What's the portion of this situation? With TERRA as morning satellite and AQUA as afternoon satellite, the diurnal variation curves could not be captured well and the time and frequency of Tmax and Tmin in each pixel/location/grid could not be monitored even using the local sine function (Mao 2016, Jiang 2010) under clear sky days. How to process the temperature data in the same location/same time with missing observation and reanalysis data? And how far is the closest location? (0.3°) Tave in (Mao 2016) is only limited in "daily mean temperature in various times (1:30, 10:30, 13:30, 22:30)" as clear shown in that paper from remote sensing data (it is feasible under the definition of Tm = (Tm 1:30+Tm 10:30+Tm 13:30+Tm 22:30)/4). The intervals between these 4 MODIS crossing time are 9 hours, 3 hours, 9 hours and 3 hours respectively. But the concept of Tave in this study should be the same with that from meteorological stations, it should be continuous in a day. Besides, the local sine function from (Mao 2016) was obtained for the global from MODIS data during 2001-2012, then whether the equation 5a-6b is statistically computed for China during 2002-

2018 to get the coefficiencies? The new simulation method for the diurnal variation of temperature - sub-sine simulation in (Jiang 2010) was based on the hypothesis that the time of Tmax/Tmin is 12 hours later than the time of Tmin/Tmax and the variation from Tmax/Tmin to Tmin/Tmax changes sinusoidal but the temperature in China during 1979-2018/2002-2018 is also this case?

**Response 4: Thank you for your guidance. You are right, and it is relatively more difficult under cloudy and complicated weather conditions. In fact, whether it is sunny or cloudy or other complicated conditions, the daily temperature changes partially satisfy the sine or cosine function changes. We already have data with a daily interval of 3 hours, so we can determine the time range for the maximum and minimum values. But in order to improve the accuracy, we further constructed the local cosine or sine function and used the daily hourly data of ERA5 as auxiliary data, which can already ensure that we can calculate the local maximum and minimum accurately. In order to calculate the maximum or minimum value, we only need the three locally adjacent maximum or minimum values to determine the coefficients of the function. In this research, we recalculate the coefficients for each pixel, because the maximum and minimum values of places with different latitude, longitude, and altitude are different and change with time. Mao et al. (2016) did a global temperature analysis, and they analyzed the global as a point (a value). In their research, there is no need to distinguish the weather state, so their coefficients can be unchanged. For our study, the weather is changed, so the coefficients of different pixels are changing, and even the same pixel is changing with time. We have made more supplementary explanations in the manuscript.**

**Point 5:** Not match between "weather status" (like fog, rain, snow) and M*D11 QC fields. Please consider the description "weather status" in the paper. Weather status records observed by observers in meteorological station are not used in this study, as introduced in "3. Data".

**Response 5: Thank you for your guidance. We have made revisions to the manuscript. The weather conditions we are referring to here mainly distinguish between clear and non-clear sky. Because the data of weather records in many stations are inconsistent and some data are missing, and there are no weather stations in most areas. So here we mainly use mature MODIS products for judgment. In addition, in fact, we already have 3 hourly interval data every day and ERA5 low-resolution hourly data. These data are also the main basis for our judgment. We have made supplementary explanations in the manuscript.**

**Point 6:** For EAR5 and CMFD datasets, the data themselves are relatively systematic. 2m air temperature from EAR5 reanalysis data is not suitable for evaluation separately like this as truth/reference values for it is the atmospheric reanalysis product, and the

advantage, the main application are in isobaric pressure levels while the parameters near the surface is suggested to be analyzed systematically and integrally. The conclusion in Line 200 "the temperature data in the ERA5 data is selected to reconstruct the Ta dataset" may not be feasible. And, why the hourly ERA5 data is suitable to determine the daily weather status in each grid?

**Response 6: Thank you for your guidance. EAR5 is only used for auxiliary data, which is used to determine the changing trend of temperature and help us determine the time range of the maximum and minimum temperature, and thus determine the coefficient of the function. It is not used to judge the weather status, and we have made revisions.**

**Point 7:** Temperatures are comprehensively influenced by solar short-wave radiation, surface long-wave radiation of the earth, atmospheric water cycle, the weather process, etc. Comparing with other atmospheric elements, its variation extent is relatively small. Sometimes even research method is not perfect but the bias of the temperature is still in a small range. So, it is expected that the correlation analysis results are not poor.

**Response 7: Thank you for your guidance. The trends are pretty good.**

**Point 8:** Why the different colors of scattered dots or anomaly bars used in Figure 6-13?

**Response 8: Thank you for your good suggestion. The scatter points of different colors in the figure are used to distinguish the accuracy values before and after correction. In Figure 9-11, the black scatter points are before correction, and orange, blue and green represent the corrected Tmax, Tmin, and Tavg respectively. It can be seen from the figure that the accuracy of the temperature (Tmax, Tmin, and Tavg) after the further correction has been greatly improved.**

**Tiny problems.**

(1) Line 80, it is suggested to consider the classification again or not mention there are 5 categories of estimation methods from remote sensing data.
**Response: Thank you for your good suggestion, and we have made revisions.**

(2) Figure 2, Quality control file exist, is it MODIS LST QC field exist? "meteorological station data", revise the description. The equations are too small to read in Figure 4.
**Response: Thank you for your good suggestion. It is the MODIS LST QC field, and we have made revisions.**

(3) Some mistakes are necessary to revise, the following are for your reference.

Line 57, "cold days and nights shorten", are they "number of days"?
Line 68, time resolution may be temporal resolution.
Line 69, which, is it "This way of detection"?
Line 70, add "relatively".
Line 102, the citation should be NCEP but not JRA-55. It is Kalnay but not Kobayashi.
Line 109, forcing datasets, it is suggested not to use it directly. Such as, there are meteorological forcing dataset, atmospheric forcing dataset, and precipitation forcing dataset.
Line 188, absorb, not suitable here, is it assimilate, or apply?
Line 232, MODIS provides QC fields, the subject of the sentence is wrong, MODIS is a sensor. Line 374 and 711, the citation time of the paper is wrong for it is Ge 2014 but not Ge 2015. Line 640, add "data", add "to", the emphasis here should be "with geostationary satellite DATA, the variation could be monitored".
5.2 title, it is not calibration. The term is not suitable here.
**Response: Thank you for your good suggestion, and we have made revisions.**

(4) Some descriptions are difficult to understand.
Line 177-178, it is suggested to revise the description, "Many studies and analyses show that the dataset's accuracy is high enough to meet the application requirements", what aspects of the accuracy and what kind of application?
**Response: Thank you for your good suggestion, and we have made revisions. Some studies use CMFD temperature data as input parameters to construct a surface air temperature model, which shows that the correlation coefficient between CMFD temperature and measured data is greater than 0.99 and has high consistency, and grid data can reflect the temporal and spatial changes of regional air temperature (Zhang et al., 2019; Wang et al., 2017). The CMFD data as an input element to build a surface temperature model can also significantly reduce model deviation and improve model accuracy (Chen et al., 2011).**

Line 274, "Clouds and water vapor have a great influence on visible light and thermal infrared remote sensing", it is radiation but not light, and the emphasis is observation. The sentence should be rewritten. The description of the paragraph following Figure 3 is not clear, especially line 306-311. What's the distance from the closest adjacent stations used for missing gaps? If it is far, the method is not right.
**Response: Thank you for your guidance. We have made revisions to the manuscript and redraw Figure 3. You are right. In fact, we deal with missing data in accordance with this principle. We mainly use 0.3° as a judgment condition. When the distance between stations is less than 0.3° and there are sites, we use site data. When the distance between stations is less than 0.3° and there is no site data, we use the interpolation method. When the distance between sites is greater than 0.3°, we downscale using ERA5 and CMFD data as well as MODIS data.**

Line 391-392, the descriptions are not suitable, "we needed to consider the degree of influence of cloudy-sky weather phenomena. First, we performed effective value statistics on the MODIS data.", "When not all pixels of the MODIS data were valid".

**Response: Thank you for your guidance, and we have made revisions. Here we mainly want to make full use of the advantages of various data, especially with the help of high-resolution MODIS data to improve the accuracy of the dataset as much as possible. According to the QC field of MODIS data, when the quality of MODIS data is guaranteed, we use MODIS data with high spatio-temporal resolution to improve local accuracy.**

(5) https://doi.org/10.5281/zenodo.5513811 is the dataset but not Model code and software.

**Response: Thank you for your good suggestion. The model code is at the bottom of the link page, and the folder is named program.zip.**

---

## Author Comment (AC4)

Dear referees,

Thank you for your valuable comments on our manuscript. First, we would like to express our sincere appreciation for your professional and insightful remarks on our paper. These comments are all valuable and have helped us to improve the quality of our paper. We have studied each comment and have made revisions that we hope will meet with approval. Please find our detailed responses below. For convenience, we also attach a version of the manuscript with changes incorporated. Thanks again.

With our best regards,
Shu Fang and co-authors
* * *
**Response to referees**

Authors presented a dataset of land surface air temperature over China using in situ station data that is interpolated to 0.1º resolution by downscaling methods using ERA5, CMFD, and CMA data. They found the data set has a reasonable accuracy quantified by RMSE, MAE, and R2, and by cross-validation method. The days of the 90th/10th percentile temperature are increasing/decreasing, which is consistent with overall warming climate. My major comments are (a) the cross-validation part (Table 1) could be extended while the evaluation against in situ data (Figures 7-9, 10-12) can be shortened, and (b) it might be helpful if the data set can be compared against other independent datasets. A major revision is recommended.
**Response: Thank you for your guidance, all the comments have been carefully and individually addressed. We have made revisions and enclosed below are our point-to-point responses to these comments.**

1. L30-31, "and we further improve data accuracy through building correction equations for different regions" => and the data accuracy is improved through building correction equations for different regions
**Response: Thank you for your good suggestion, and we have made revisions.**

2. L41, "reliable accuracy", which needs explanations. Can you justify that RMSE of 0.86-1.78ºC is a reliable accuracy?
**Response: Thank you for your guidance, and we have made revisions. We are mainly based on current application requirements and accuracy comparison with other datasets.**

3. L57-58, L58-59, The statements of "the extremely cold days and nights gradually shorten" and "intensity and duration of extreme weather events are also increasing" do not appear consistent. Are the increasing extreme events all hot events?
**Response: Thank you for your guidance, and we have made revisions. The number of extremely cold days and cold nights gradually shortens and the frequency of extreme weather events is also increasing.**

4. L66-68, "monitoring estimating/obtaining/ Ta" => Ta monitored/estimated/obtained

**Response: Thank you, and we have made revisions.**

5. L88, delete "energy"

**Response: Thank you, and we have made revisions.**

6. L135-136, "cumulative temperature is between 2500°C and 4000°C.", a citation is needed. Why the cumulative temperature is used instead of average temperature? How the temperature is accumulated? If so, the unit should be ºC*year.

**Response: Thank you for your guidance. There are two main methods for calculating the accumulated temperature. Here we mainly use the accumulation of the daily average temperature ≥10 °C. We have added references here, and the unit is also in accordance with the commonly used unit. Here is mainly to introduce the different conditions of six subregions, there is no special reason for the choice.**

7. L145, "800°C" should be a typo.

**Response: Thank you, and we have made revisions.**

8. L146-151, (V), what temperature is this region?

**Response: Thank you, and we have made revisions.**

9. Section 2, How are the boundaries of these regions determined?

**Response: We divide China into six regions mainly based on climatic conditions, such as temperature and rainfall, and topographical conditions such as elevation.**

10. L200 and L185, it is not clear whether one or both are used to reconstruct Ta. How are these data used to reconstruct Ta, as a training data or validation data?

**Response: Here we mainly want to introduce that ERA5 is an upgraded version with higher accuracy. In the non-clear sky condition, we used the ERA5 dataset for downscaling where there is no in situ data. Part of the data is used as training data, and part is used as validation data.**

11. L208-210, reference is needed such as Du et al. 2020

**Response: Thank you, and we have made revisions.**

12. L244 "In currently" delete "in"

**Response: Thank you. We have made revisions.**

13. Figure 3, in "2002-2018" branch, are there any in situ data used? If not, are the final output purely from MODIS observation?

**Response: Thank you for your guidance. During 2002-2018, we mainly used MODIS data as the judgment condition, but when MODIS data is missing, in situ**

data and other data are also used at the same time. The manuscript didn't introduce it in detail, we revised it.

14. L305, "was less than 0.3°," is questionable because the data resolution is 0.1º.
**Response: The spatial resolution of ERA5 data is 0.3°. In order to ensure the consistency of our data, we use 0.3° as the condition for judging the distance between observation stations. When there is a station at the pixel location or the station distance is less than 0.3°, we use the in situ data. When the distance between stations is greater than 0.3°, we use ERA5 data for downscaling.**

15. L306-315, Is there any elevation consideration between stations when filling from adjacent stations?
**Response: Yes, we have considered the influence of elevation. In order to improve the accuracy of the interpolation, we first correct the data of the observation site to a uniform sea level and then perform further calculations based on the elevation of the interpolation point to obtain the corresponding temperature.**

16. L323-326, again, any elevation consideration?
**Response: Yes, we have considered the influence of elevation and have made revisions in the manuscript.**

17. Equations (1)-(2), a reference is needed. Why do the indices I and j start from 0?
**Response: Thank you for your guidance. This formula is derived by ourselves and does not require references, and we have made revisions.**

18. In Figure 4, the 3rd column, the downscaling should use the surrounding data, not from all from the corner (The figure is merely visible due to low quality/resolution)
**Response: Thank you for your guidance. We have made revisions to the manuscript and redraw Figure 4.**

19. L369-370, How are At and Bt "obtained", by least square method in the next sentence?
**Response: Thank you for your guidance. $A_t$ and $B_t$ are obtained by the least square method. We didn't introduce it in detail, and we have modified it in the manuscript.**

20. L403-404, Should "use … as" be "average" or "to calculate daily average temperature"?
**Response: Thank you for your guidance, and we have made revisions.**

21. L427. "Sect. 0,", please check.
**Response: Thank you, and we have made revisions.**

22. L451-453, Any reason for "we selected areas with uniform surface types and flat terrain under clear skies as the comparative study area and compared this product with

the existing datasets"?

**Response: Thank you for your guidance. It is very difficult to validate large-scale data. First of all, it is necessary to ensure that the reference dataset is relatively accurate. We mainly consider the representativeness of the in situ data from the ground station, so we choose as much as possible the area with a single surface type and flat terrain for comparison. Currently, there is no better method for large-scale calibration.**

23. L455, Since ERA5 and CMA data have been used in the downscaling processes, they are not independent anymore and it is questionable to be used for validation/evaluation.

**Response: Only ERA5 and CMFD data during non-clear sky are used for downscaling and filling. When doing validation, we do cross-comparison after upscaling all data to the same resolution, including ERA5 and CMFD data that did not participate in the calculation. The most important thing is that we made a cross-comparison with the CMA data that has not been involved in the calculation, and used the site data that did not participate in the calculation for validation.**

24. L466, Should TXx/TNn be better TXn/TMn? Otherwise, what are the meaning of "x" and "N"?

**Response: Here we mainly use the definition of World Meteorological Organization (WMO) [1][2]. WMO mainly defines 27 core indices, and we quote them in their original form here. TXx/TNn respectively represent the maximum value of maximum temperatures and the minimum value of minimum temperatures. The x in TXx represents max, and the n in TNn represents min in the same way.**

**1.Karl, T.R., N. Nicholls, and A. Ghazi, 1999: CLIVAR/GCOS/WMO workshop on indices and indicators for climate extremes: Workshop summary. Climatic Change, 42, 3-7.**

**2.Peterson, T.C., and Coauthors: Report on the Activities of the Working Group on Climate Change Detection and Related Rapporteurs 1998-2001. WMO, Rep. WCDMP-47, WMO-TD 1071, Geneve, Switzerland, 143pp.**

25. Figures 6-8, Since the "dataset" have been using "in situ data" at pixels where in situ data exist, I doubt these comparisons are meaningful. A better way is to compare independent data/station, or compare with other available datasets created independently.

**Response: Maybe it was not clearly written in the manuscript, and we revised it. In order to ensure the independence of the data, we use the jackknife method to divide the in situ data into 80% and 20%. Of these, 80% is used to build the model, and 20% is used to validate accuracy. Figures 6-8 and Figures 9-11 are accuracy validation under the premise of 20% validation data.**

26. Figures 9-11, some comments as for Figure 6-8.

**Response: Thank you for your guidance, and we have made revisions.**

27. Table 1, Glad to see the cross validation. I would suggest replot Figures 7-9 using independent data and put original figure to appendix or supplementary materials. The question is how the cross-validation is arranged, which may need explanations in text.

**Response: Thank you for your guidance. We used independent data and in situ data for cross-validation and validation. We left part of the data and did not participate in the calculation, so the validation is no problem. The expression may not be clear enough, we have revised it in the manuscript.**

28. L473, "with more than 90% (less than 10%) correlation with the number of days in each year" may need to rephrase or rewritten. It is hard to understand. How do these percentile is determined, based on a certain time period?

**Response: Thank you, and we have made revisions. We select TX90p and TN10p as extreme climate indices defined by the WMO organization [1][2]. The 90% (10%) corresponding value in the time series is used as the threshold for judging warm days (cold nights).**

**1.Karl, T.R., N. Nicholls, and A. Ghazi, 1999: CLIVAR/GCOS/WMO workshop on indices and indicators for climate extremes: Workshop summary. Climatic Change, 42, 3-7.**

**2.Peterson, T.C., and Coauthors: Report on the Activities of the Working Group on Climate Change Detection and Related Rapporteurs 1998-2001. WMO, Rep. WCDMP-47, WMO-TD 1071, Geneve, Switzerland, 143pp.**

29. Figure 14c', check x-axis, any ***?

**Response: Thank you, and we have made revisions and redrawn Figure 14.**

---

## Author Comment (AC5)

**Response to Chief Editors**

Discussion among Chief Editors:

Question: This paper (essd-2021-309) describes a high-resolution dataset of near-surface air temperature in China from 1979 to 2018.

I am wondering if it is fair to ask them to update the time series, for example, to 2020? Nearly 3-year data latency seems to be too long for air temperature data in this day and age.

Feedback: Without reading the paper or knowing more about their sources, I don't know their options. I do know that, in the world of global data products, sometimes 2018 represents the most recent publicly available quality-controlled product.

In global emissions, mostly due to tardy national reporting, we have - for key GHG - nothing more recent than 2018. Some forest inventories update only at five-year intervals (with some countries still lagging). In ESSD-221-228, the global GHG inventory paper prepared for IPCC WG III and for CoP26, they report only through 2018 then extrapolate to 2019. Carbon budget does the same; report through two years earlier then extrapolate to cover the most recent year. E.g. 2021 version of carbon budget (delayed this year by nearly two months), appearing in late 2021, will report through 2020 with caveats while extrapolating for 2021 based on initial months and preliminary estimates. Real-time (daily) reporting from NOAA of Moana Loa $CO_2$ proves so important, but even our friends at ERL produce quality-controlled reports only after a couple months of checking? Tricky business during pandemic disruptions. Real-time crowd-sourced aviation data, for all its other weaknesses, all of a sudden assumes remarkable relevance?

For air temperature, one can find monthly reports with only a month lag from some sources but a full accurate quality-controlled annual product sometimes requires as much as six months processing before release? Longer if one wants merged surface and satellite data? If authors submitted the paper elsewhere then did not update it in the interim, they could easily have included only data through 2018 in their original manuscript. Or it came from a thesis but student then took a year or more to extract it for publication? For small changes, e.g. numeric updates, we allow final updates at proof stage.

Good question, not always the most obvious answer. Ask gently ...

**Response: Thank you very much for Chief Editors' attention to our dataset and manuscripts. Our China National Key R&D Program support ends at the end of this year (2021), and the time for hourly data sharing agreement with the China Meteorological Administration is up, and the data-sharing agreement stipulates that the data was only provided until 2018. Especially for the hourly data of the past 3 years, there are now new regulations. Until there is no new cooperation project support, a large amount of station hourly data across China will not be provided to us in the short term. Without the support of ground weather observation station hourly data across China, it is difficult to guarantee the completion of this work. Of course, once we can obtain the relevant data, we will**

continue to complete the work and update the dataset.

In addition, we are studying the use of MODIS remote sensing data to retrieve the surface air temperature with a resolution of 1 km. The theoretical accuracy of the algorithm is very good, but the actual application will take some time. Currently, we are improving the algorithm. In 2022, we will complete a 1-km near-surface air temperature dataset using remote sensing data from 2001 to 2021. Thank Chief Editors and reviewers for your support. Our dataset will be submitted to ESSD first. At that time, we will also share the dataset and invite you for guidance.